# Accurate and Diverse LLM Mathematical Reasoning via Automated PRM-Guided GFlowNets

## Abstract

Achieving both accuracy and diverse reasoning remains challenging for Large Language Models (LLMs) in complex domains like mathematics. A key bottleneck is evaluating intermediate reasoning steps to guide generation without costly human annotations. To address this, we first introduce a novel Process Reward Model (PRM) trained automatically using Monte Carlo Tree Search coupled with a similarity-based data augmentation technique, effectively capturing step-level reasoning quality. Leveraging this PRM, we then adapt Generative Flow Networks (GFlowNets) to operate at the reasoning step level. Unlike traditional reinforcement learning focused on maximizing a single reward, GFlowNets naturally sample diverse, high-quality solutions proportional to their rewards, as measured by our PRM. Empirical evaluation shows strong improvements in both accuracy and solution diversity on challenging mathematical benchmarks (e.g., +2.59% absolute accuracy on MATH Level 5 for Llama3.2-3B), with effective generalization to unseen datasets (+9.4% absolute on SAT MATH). Furthermore, we benchmark our PRM against existing open-source reward models, demonstrating superior alignment with reasoning quality and more consistent guidance for downstream generation. Our work demonstrates the potential of PRM-guided, step-level GFlowNets for developing more robust and versatile mathematical reasoning in LLMs.

## 1 Introduction

Large Language Models (LLMs) have demonstrated remarkable progress in various natural language tasks (Brown et al., 2020; Dubey et al., 2024), yet achieving robust and reliable mathematical reasoning remains a significant challenge (Lewkowycz et al., 2022; Hendrycks et al., 2021). While LLMs have shown increasing proficiency in solving mathematical problems (Cobbe et al., 2021; Yuan et al., 2023), current approaches often prioritize accuracy on benchmark datasets (Hendrycks et al., 2021; Wang et al., 2022), potentially overlooking other crucial aspects of intelligent reasoning, such as the ability to explore and generate diverse solution strategies (Naik et al., 2023). For LLMs to truly excel in mathematical domains and move beyond pattern recognition towards genuine understanding, they must not only arrive at correct answers but also exhibit the capacity to reason through problems in multiple, varied, and insightful ways (Yu et al., 2024; Uesato et al., 2022).

Traditional reinforcement learning methods like Proximal Policy Optimization (PPO; Schulman et al., 2017) have shown promise in improving LLM mathematical reasoning (Yao et al., 2023). However, these methods inherently aim to maximize a single reward signal, often leading to the exploitation of a narrow set of solution strategies (Ziegler et al., 2019; Naik et al., 2023). This limitation becomes particularly critical when considering the development of robust and generally applicable problem-solving AI systems, where adaptability to novel situations and the exploration of diverse solution spaces are paramount.

Addressing the need for diverse reasoning requires effective guidance at the level of intermediate steps. However, obtaining reliable reward signals for these steps typically involves expensive human annotation. We overcome this limitation by first developing an automatically trained Process Reward Model (PRM; Uesato et al., 2022). Our approach uniquely employs Monte Carlo Tree Search (MCTS; Luo et al., 2024) combined with a novel similarity-based data augmentation technique to

generate high-quality step-level reward signals without manual labeling. Given this automated and nuanced step-level reward mechanism provided by the PRM, we then propose leveraging Generative Flow Networks (GFlowNets; Bengio et al., 2021; 2023) for fine-tuning. Unlike traditional reinforcement learning methods that often collapse to a single strategy, GFlowNets are designed to sample proportionally to rewards, naturally fostering solution diversity. Our key adaptation is operating GFlowNets at the reasoning step level rather than the token level – each state represents a partial solution, and actions generate complete reasoning steps – allowing the PRM reward to guide the exploration of diverse, high-quality reasoning paths effectively. Prior work using GFlowNets for LLM fine-tuning (Hu et al., 2023; Takase et al., 2024) often utilized variants of the Subtrajectory Balance (SubTB) loss (Madan et al., 2023), which we adapt for our step-level framework.

Our main contributions are:

- **Automated Process Reward Model**: An efficient training methodology featuring an automatically trained PRM using Monte Carlo Tree Search and novel similarity-based data augmentation, eliminating the need for expensive human step-level annotations while achieving superior data efficiency through rollout reuse.

- **Step-Level GFlowNet Framework**: An adaptation of GFlowNets to operate at complete reasoning steps rather than individual tokens, providing semantic coherence and enabling fine-grained quality control through step-wise PRM evaluation. This addresses key limitations of existing token-level approaches.

- **Strong Empirical Validation**: Demonstrated improvements in both accuracy and solution diversity across challenging benchmarks, with particularly impressive generalization (+9.4% absolute on SAT MATH for 3B model), indicating that step-level diversity training captures transferable reasoning patterns.

Empirically, we demonstrate that our approach not only improves accuracy on challenging mathematical reasoning benchmarks but also generates more diverse solution strategies compared to both baseline models and PPO-fine-tuned variants. This is particularly evident in our diversity analysis, where GFlowNet-fine-tuned models show significantly lower semantic similarity between generated solutions while maintaining correctness.

Our work points towards new directions for developing next-generation LLMs with enhanced reasoning capabilities. By demonstrating a method to improve both accuracy and diversity in a complex domain like mathematical reasoning, we open possibilities for creating more robust, versatile, and ultimately more intelligent LLM systems capable of tackling a wider range of challenging problems.

## 2 BACKGROUND AND RELATED WORK

**Mathematical Reasoning in LLMs** has seen significant progress through various approaches, including chain-of-thought prompting (Wei et al., 2022), self-consistency (Wang et al., 2022), and reward modeling (Lightman et al., 2023; Zhang et al., 2024). Process reward modeling, pioneered by Lightman et al. (2023), has evolved to include automated approaches: Zhang et al. (2024) introduced ReST-MCTS* for self-training via process reward guided tree search, while Guan et al. (2025) demonstrated that small language models can achieve state-of-the-art mathematical reasoning using MCTS-guided process rewards.

While these methods have improved accuracy, they often lack mechanisms for promoting solution diversity. Recent work by Wang et al. (2024b) highlights the importance of diverse reasoning paths but focuses primarily on accuracy rather than explicitly encouraging diversity during training.

Chain-of-thought (CoT) approaches (Wei et al., 2022) and their variants such as Tree-of-Thought (Yao et al., 2024) and Graph-of-Thought (Besta et al., 2024) have demonstrated substantial improvements by encouraging models to articulate intermediate steps in their problem-solving process. However, the precise mechanisms underlying these improvements remain an active area of investigation, with some researchers questioning whether the benefits derive specifically from human-like task decomposition or simply from the additional computation afforded by generating more tokens (Pfau et al., 2024). Several enhancements to CoT approaches have been proposed, including using datasets of preference pairs of reasoning traces to finetune the CoT-generating model (Lahlou et al., 2024).

Despite these advances, current approaches to mathematical reasoning predominantly emphasize accuracy improvements rather than fostering diverse solution strategies. This limitation becomes particularly relevant in educational contexts and mathematical exploration, where multiple valid approaches can provide deeper insights into problem structures. This gap motivates our investigation into leveraging generative flow networks to promote both accuracy and diversity in mathematical reasoning.

**Generative Flow Networks (GFlowNets)** represent a novel framework for learning to sample from a desired distribution, offering advantages over traditional reinforcement learning approaches (Bengio et al., 2021; 2023). GFlowNets operate on a directed acyclic graph (DAG) structure, where states $\mathcal{S}$ represent partial constructions and actions $\mathcal{A}$ represent transitions between states. This graph contains a unique source state $s_0$ with no parents and a sink state $s_f$ with no children. States that connect directly to $s_f$ are termed terminal states $\mathcal{X}$, each associated with a positive reward $R(x) > 0$ for $x \in \mathcal{X}$.

The core objective of GFlowNets is to learn policies that generate complete trajectories $\tau = (s_0, s_1, ..., s_n, s_f)$ such that terminal states are sampled with probabilities proportional to their rewards: $P(x) \propto R(x)$. This is achieved through flow conservation and reward matching constraints that ensure the learned policy samples diverse, high-reward solutions rather than converging to a single optimal path.

Unlike reinforcement learning methods that focus on maximizing expected cumulative rewards - typically converging to deterministic policies that exploit highest-reward paths - GFlowNets learn stochastic policies that maintain diversity while still favoring high-reward solutions. This balance between exploration and exploitation makes GFlowNets particularly valuable for applications where multiple viable solutions are preferable to a single optimal one.

Various training objectives have been proposed, including trajectory-balance (TB), detailed-balance (DB), and Subtrajectory Balance (SubTB) methods. Our work builds on SubTB($\lambda$) (Madan et al., 2023), which enables learning from incomplete trajectories - particularly suitable for our step-level framework.

GFlowNets have demonstrated remarkable success across diverse domains. Their ability to generate diverse, high-quality samples addresses a fundamental challenge in scientific discovery tasks involving astronomically large search spaces. By learning to sample diverse high-reward candidates, GFlowNets can efficiently identify promising regions of the design space while maintaining the variety needed to accommodate additional constraints not captured in the primary reward function, such as synthesis feasibility or absence of side effects. Recent work has explored GFlowNets for language generation tasks (Hu et al., 2023; Takase et al., 2024; Ho et al., 2024). Most notably, Takase et al. (2024) demonstrated that GFlowNet fine-tuning can generate diverse correct solutions for mathematical reasoning tasks. However, their approach operates at the token level, with states and actions defined over individual tokens: $\pi_\theta(Y|X) = \prod_i \pi_\theta(y_i|X, y_{1:i-1})$. In contrast, our work operates at the reasoning step level, where each action corresponds to generating a complete reasoning step rather than individual tokens. This step-level granularity enables more semantically meaningful control over solution generation while maintaining the diversity benefits of GFlowNets.

The step-level approach offers several advantages over token-level methods like Takase et al. (2024): (1) **Semantic Coherence**: Each action generates a complete, meaningful reasoning step rather than individual tokens, avoiding the semantic fragmentation that can occur in token-level generation; (2) **Reward Alignment**: Step-wise PRM evaluation provides more accurate reward signals than token-level approaches, where individual tokens may not reflect reasoning quality; (3) **Logical Structure**: Operating at the step level naturally captures the hierarchical structure of mathematical reasoning, where each step represents a logically complete inference; (4) **Quality Control**: The step-level granularity enables fine-grained control over reasoning quality while maintaining coherence within each step. This is particularly important for mathematical reasoning, where partial steps or token sequences may be mathematically meaningless, but complete reasoning steps represent verifiable logical progressions.

We discuss other related work in Appendix C. The relationship between our approach and Maximum Entropy RL is discussed in Appendix D.

# 3 PROCESS REWARD MODEL FOR MATHEMATICAL REASONING STEPS

We train a PRM $U(s'|s)$ that evaluates the quality of a proposed reasoning step $s'$ given its predecessor $s$. The PRM outputs a score between 0 and 1, representing the probability that step $s'$ will lead to a correct solution, assuming $s$ is correct.

**Relationship to Prior PRM Work**: Process reward modeling has evolved significantly since Lightman et al. (2023) introduced step-by-step verification with human annotations. Recent automated approaches include Zhang et al. (2024), who use MCTS to infer process rewards by estimating the probability that each step leads to correct answers, and Guan et al. (2025), who employ MCTS-guided process reward models for small language models.

Our approach differs in several key aspects: (1) **Data Augmentation**: Unlike prior work that discards most generated rollouts, our similarity-based augmentation leverages nearly every rollout, significantly increasing data efficiency; (2) **Continuous Scoring**: We use continuous scores rather than binary labels (Luo et al., 2024; Lightman et al., 2023; Wang et al., 2024b), enabling more nuanced evaluation of step quality; (3) **Integration**: Our PRM provides reward signals compatible with various training methods such as PPO and GFlowNets. It is particularly well-suited for step-level GFlowNet training, requiring calibrated probability scores rather than just binary scores.

## 3.1 AUTOMATED DATA GENERATION VIA MCTS

To train our PRM without human annotations, we adapt the MCTS-based data generation approach of Luo et al. (2024) with several key modifications. Following their framework, we perform Monte Carlo rollouts to evaluate step quality, but critically extend their binary scoring (correct/incorrect) to continuous values in $[0, 1]$ to capture nuanced step quality gradations.

For each candidate step $s$, we estimate its Monte Carlo value $MC(s)$ by performing $k = 96$ rollouts with temperature 0.6. Unlike binary evaluation schemes, our continuous scoring allows the PRM to learn fine-grained quality distinctions between reasoning steps: $MC(s) = $ (successful rollouts from prefix ending at step $s$)/$k$.

We use binary search to efficiently identify the first incorrect step within candidate reasoning paths. Only steps preceding the first error are included in the training dataset, as subsequent steps build upon invalid reasoning. Therefore as illustrated in Figure 3, the PRM consistently assumes that all preceding steps are correct when evaluating the current step. In other words, the PRM effectively answers the question: *to what extent can this step lead towards the correct solution, under the assumption that all prior steps are valid?* The key advantage of this approach is that, when the PRM is employed to provide a reward signal during PPO or GFlowNets training, it assigns rewards solely based on the quality of the step provided as input to the PRM. Consequently, earlier mistakes do not propagate to unduly penalize the model throughout the trajectory.

We applied this methodology using Qwen2.5-Math on 70,000 problems from OpenMathInstruct2 (Toshniwal et al., 2024). Complete algorithmic details and implementation specifics are provided in Algorithm 1 (Appendix E.1).

## 3.2 DATASET AUGMENTATION VIA ROLLOUT REUSE AND SIMILARITY GROUPING

To improve the efficiency of our data generation and create a larger, more comprehensive dataset, we implement a dataset augmentation strategy. This strategy leverages the rollouts generated during the MCTS process and incorporates a step similarity grouping technique.

During MCTS, for each evaluated step, we store the $k$ rollouts used to estimate its $MC(s)$ value. Each rollout is stored as a tuple $(r, x)$, where $r$ is the generated reasoning path (rollout) and $x$ is a binary indicator (1 or 0) denoting whether the rollout led to a correct final answer. To augment our dataset, we extract individual steps from these stored rollouts. For each rollout step, we create a new data entry. The "prefix" for this new entry is constructed by concatenating the original prefix of the step from which the rollouts were generated with the step itself. This process allows us to reuse steps from successful and unsuccessful rollouts, significantly increasing the size of our dataset. Ideally, for each step evaluated in MCTS, we could add up to $k$ new step examples to our dataset through this rollout reuse.

However, directly adding all rollout steps without further processing can introduce inconsistencies. As illustrated in Table 1, steps that contribute equivalently to the reasoning process can receive different $MC(s)$ values if evaluated in isolation with a limited number of rollouts ($k = 1$, which would be the case if we directly evaluated each step in a rollout independently). This is because evaluating with $k = 1$ can lead to noisy and unreliable step value estimations.

To address this consistency issue and further refine our dataset, we introduce a post-processing step based on step similarity grouping. First, we define a **Step Similarity Function** to quantify the similarity between two steps of mathematical reasoning. This function evaluates similarity based on two primary criteria: (1) Calculation Consistency: If both steps contain mathematical calculations, the function checks if the results of these calculations are identical. If the results differ, the steps are considered dissimilar, and the function returns a similarity score of 0. (2) Textual Similarity: If the calculation results are the same, or if neither step contains calculations, the function computes the Levenshtein distance between the textual content of the two steps.

Table 1: Example of Similar Steps with Different Values

| Steps | value |
|---|---|
| Max attended 40 college courses in 2 years | 0 |
| Within 2 years, Max enrolled in 40 college courses. | 1 |

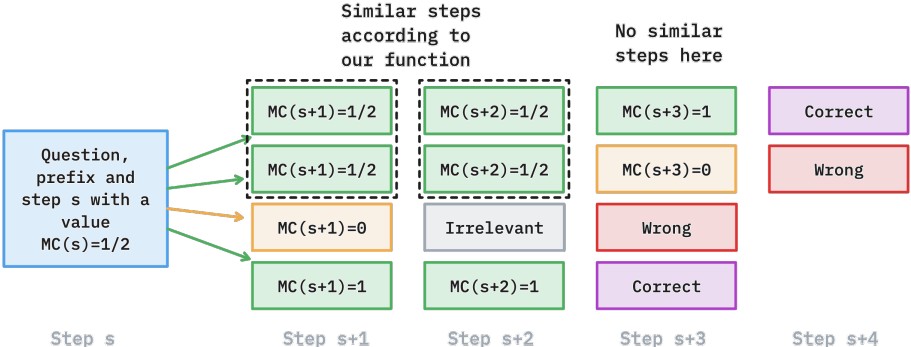

Figure 1: Data processing workflow for PRM training. Starting from a step $s$ with $MC(s) = 1/2$, the diagram shows how subsequent steps are processed based on their Monte Carlo values. Similar steps (indicated by dashed boxes) share MC values. Steps following an incorrect step ($MC = 0$) are excluded from the training dataset, as they would be built upon invalid reasoning. Gray boxes indicate steps that become irrelevant to the training process.

If the similarity score computed by the *Step Similarity Function* exceeds a predefined threshold (set to 0.85 in our experiments), the steps are grouped into the same similarity group. Within each group, we then assign consistent step values. Specifically:

- If all steps within a group originated from rollouts that led to correct (resp. incorrect) final answers, all steps in the group are assigned a value of 1 (resp. 0).

- If there is a mixture of correct and incorrect rollouts associated with the steps in a group, all steps in the group are assigned the $MC(s)$ value that was originally estimated for the MCTS step from which these rollouts were generated, as a **reasonable approximation** of the value for all similar steps in the group, assuming that steps similar to a step with a known $MC(s)$ are likely to have a similar probability of leading to a correct solution.

This combined dataset augmentation process ensures greater consistency and reliability in the step-level labels. The initial MCTS generation yielded approximately 100k step examples from the input problems described in Section 3.1. Applying rollout reuse and similarity grouping significantly expanded this set to the final 2.1M entries used for PRM training, greatly enhancing dataset size and diversity and improving the PRM's generalization capability. This final dataset comprises approximately 30% false steps, 20% steps guaranteed correct (value 1), and 50% intermediate steps (value between 0 and 1). The complete data processing workflow, including the handling of incorrect steps during MCTS and the augmentation logic, is visually summarized in Figure 1.

### 3.3 PRM TRAINING

We fine-tuned Qwen2.5-Math (Qwen et al., 2025) as our PRM. We train the PRM to predict the probability of a step leading to a correct solution using Binary Cross-Entropy Loss (BCELoss) $L = -\frac{1}{N} \sum_{i=1}^{N} [y_i \log(\hat{y}_i) + (1 - y_i) \log(1 - \hat{y}_i)]$, where $y_i$ is the true label for the $i$-th step (i.e., the $MC(s)$ value), and $\hat{y}_i$ is the predicted probability for the $i$-th step, given by $\hat{y}_i = \mathrm{PRM}(\mathrm{question}, \mathrm{solution\_prefix}, \mathrm{step})$.

### 3.4 PRM VALIDATION

We validate our PRM through three complementary analyses. First, we assess its ability to detect step-level errors by corrupting correct solution steps in systematic ways (e.g., changing numbers, or removing key reasoning components). The PRM consistently assigns lower scores to corrupted steps, demonstrating its sensitivity to reasoning errors (See Appendix E.4).

Second, we evaluate its capability to support diverse solution paths by comparing scores assigned to different valid approaches for the same problem (using the GSM8K dataset). The PRM assigns comparable scores to different valid approaches, indicating its ability to recognize multiple correct reasoning paths (See Figure 3 in Appendix E.5).

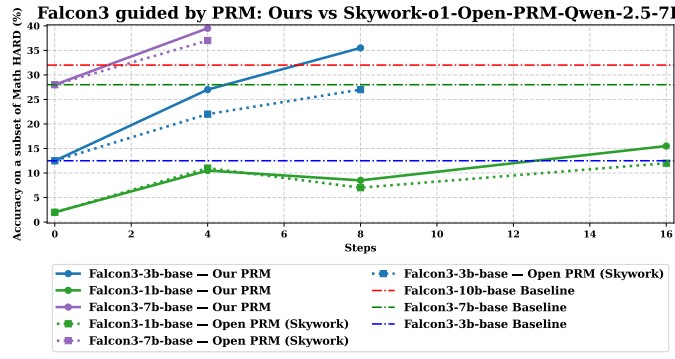

Figure 2: Accuracy of Falcon models on a subset of MATH Hard using PRM-guided search with varying numbers of proposed steps $k$. Horizontal lines indicate the baseline accuracy of unguided Falcon3-3B, 7B, and 10B models using prompt-based decoding. Solid (resp. dotted) curves represent the accuracy of Falcon3-1B, 3B, and 7B guided by our PRM (resp. Skywork-o1-Open-PRM-Qwen-2.5-7B).

Thirdly, we implemented a guided search strategy to enhance the mathematical reasoning capabilities of LLMs using our trained PRM, similar to the approach in Snell et al. (2024). At each step of generation, the LLM proposes $k$ candidate steps (generated with a temperature of 0.8 for diversity), and the PRM selects the step with the highest score. This step is then appended to the prompt, and the process is repeated until a complete solution is generated.

Figure 2 demonstrates the effectiveness of PRM-guided search on a subset of the MATH Hard dataset. These results highlight the value of PRMs in guiding step-by-step reasoning. By selecting the most promising steps according to the PRM's evaluation, the guided search strategy substantially enhances the accuracy of smaller models, in some cases enabling them to approach or even surpass the performance of larger unguided models on challenging mathematical reasoning tasks. Similar applications of PRMs can be found in Wang et al. (2024a;b), as well as in related "helper" models such as the Preference Process Model in Guan et al. (2025). Additional tests of the PRM are provided in Appendix E.4

## 4 STEP-LEVEL GFLOWNET FINE-TUNING FOR DIVERSE SOLUTIONS

### 4.1 STEP-LEVEL GFLOWNET FRAMEWORK

Building on Hu et al. (2023), we adapt GFlowNets to operate at the reasoning step level rather than the token level. Our key insight is that mathematical reasoning has a natural hierarchical structure where complete reasoning steps represent semantically meaningful units.

**State and Action Space**: A state $s$ represents a partial solution consisting of the question and all generated reasoning steps up to that point. An action corresponds to generating a complete reasoning

step, transitioning from state $s$ to state $s'$. This step-level granularity enables fine-grained control over the reasoning process while maintaining semantic coherence within each step.

**Reward Structure**: The reward $R(s_n)$ for a complete reasoning trajectory is computed using our PRM scores $R(s_n) = \prod_{i=1}^{n} U(s_i|s_{i-1})$, where $U(s_i|s_{i-1})$ is the PRM score for step $s_i$ given the previous partial solution $s_{i-1}$. This multiplicative structure ensures that trajectories with any low-quality steps receive low rewards, encouraging high-quality reasoning throughout. We propose in Appendix F a **Bayesian interpretation** of our approach.

**Policy Parameterization**: The forward policy $\pi(s'|s)$ is parameterized by an LLM that generates the next reasoning step. Following Bengio et al. (2023), we incorporate a sink state $s_f$ to handle variable-length solutions, with $\pi(s_f|s)$ representing the termination probability.

## 4.2 Training Objective and Implementation

We adapt the Subtrajectory Balance (SubTB) loss (Madan et al., 2023) for our step-level GFlowNet framework, following Hu et al. (2023) who were the first to use GFlowNets for LLM fine-tuning and demonstrated the effectiveness of SubTB for language generation tasks.

**Mathematical Formulation**: The SubTB loss for our step-level framework is:

$$\mathcal{L} = \sum_{0 \le i < j \le n} \lambda^{j-i} \left( \log \frac{R(s_i) \prod_{k=i+1}^{j} \pi(s_k|s_{k-1})\pi(s_f|s_j)}{R(s_j)\pi(s_f|s_i)} \right)^2 \tag{1}$$

where $\lambda \in [0,1]$ is a discount factor, $R(s_i)$ is the PRM-based reward for partial trajectory up to step $i$, and $\pi(s_f|s_i)$ represents the termination probability. This formulation ensures that the learned policy satisfies the detailed balance condition across all subtrajectories, leading to proper reward-proportional sampling.

Our training procedure maintains a replay buffer $\mathcal{B}$ of size 1000 storing complete trajectories. The replay buffer prioritizes trajectories with higher rewards to accelerate learning. For each question, we (i) generate $k$ candidate reasoning paths using current policy, (ii) evaluate rewards for terminating states using PRM, (iii) Update policy using modified SubTB loss, (iv) Store successful trajectories in $\mathcal{B}$. Our approach is detailed in Algorithm 2 in Appendix G.2. We further implement a prioritized replay buffer, temperature scheduling during trajectory generation, and gradient stabilization. We detail these additions in Appendix G.3.

## 5 Experiments

For the main fine-tuning experiments, we used Llama3 models as the base architectures (specific sizes detailed in Appendix G.1). These models were fine-tuned using either our step-level GFlowNet approach or a PPO baseline. Both fine-tuning methods utilized reward signals from our Process Reward Model (PRM), which was pre-trained using the methodology described in Section 3 (based on the Qwen2.5-7B-math model (Qwen et al., 2025)).

**PPO Baseline Implementation**: Our PPO implementation follows standard practices adapted for sequential reasoning. For each training question, the LLM incrementally constructs its reasoning trajectory by generating one step at a time. At every stage of this sequential process, only the newly proposed step is subjected to evaluation: the PRM assesses this individual step and provides a step-level reward signal. This localized feedback guides the model throughout the reasoning process, ensuring that each successive step is informed by fine-grained evaluations rather than a single outcome-level supervision. The PPO objective maximizes expected cumulative rewards using the clipped surrogate objective with entropy regularization. We use identical model architectures, training data, and PRM evaluation as our GFlowNet approach to ensure fair comparison. PPO hyperparameters include: learning rate 5e-6, batch size 144, PPO clip ratio 0.2, value function coefficient 0.5, and entropy coefficient 0.01.

The GFlowNet and PPO fine-tuning was conducted on a challenging subset of 10,000 questions from the OpenMathInstruct2 dataset (Toshniwal et al., 2024) ("augmented math" category, aligning with MATH Level 5 difficulty). We evaluated the performance of the fine-tuned models on the MATH Hard benchmark (Hendrycks et al., 2021), the GSM8K benchmark (Cobbe et al., 2021), and

the SAT MATH dataset (Zhong et al., 2023) to assess both in-domain accuracy and generalization capabilities.

## 5.1 MAIN RESULTS

We evaluate our approach on three challenging mathematical reasoning benchmarks: MATH Level 5 (our training domain), GSM8K, and SAT MATH (generalization domains). Table 2 presents our main results, comparing GFlowNet fine-tuning against PPO and baseline models:

The empirical results validate the effectiveness of step-level GFlowNet fine-tuning across model scales and demonstrate strong generalization capabilities. The smaller Llama3.2-3B-it model shows substantial improvements with GFlowNet fine-tuning, achieving a 2.59 percentage point gain over the baseline on MATH Level 5.

**Generalization Analysis**: The most striking results appear on out-of-domain benchmarks. On SAT MATH, GFlowNet fine-tuning achieves particularly impressive

Table 2: Performance comparison on mathematical reasoning benchmarks. MATH Level 5 was used for training; GSM8K and SAT MATH evaluate generalization. Best results for each model size are **bolded**.

| Model | MATH Level 5 | SAT MATH | GSM8K |
|---|---|---|---|
| Llama3.2-3B-it | 14.46% | 65.6% | 67.8% |
| + PPO | 15.32% | 70.0% | 68.4% |
| + GFlowNet | **17.05%** | **75.0%** | **68.5%** |
| Llama3.1-8B-it | 17.96% | 81.2% | 78.1% |
| + PPO | 18.44% | 81.2% | **79.1%** |
| + GFlowNet | **18.67%** | **84.4%** | 79.0% |

gains: 9.4 percentage points improvement for the 3B model (75.0% vs 65.6% baseline) and 3.2 percentage points for the 8B model (84.4% vs 81.2% baseline). This strong generalization can be attributed to three factors: (1) **Diverse Reasoning Patterns**: By training on multiple solution approaches rather than converging to a single strategy, the model develops more robust reasoning skills that transfer across problem types; (2) **Step-Level Granularity**: Operating at the reasoning step level captures fundamental mathematical operations that generalize across different problem formats; (3) **PRM-Guided Quality**: The PRM ensures that diverse solutions maintain high logical quality, preventing the degradation often seen when optimizing for diversity alone.

The consistent improvements across both model sizes indicate that our approach scales effectively, while the superior generalization compared to PPO suggests that reward-proportional sampling captures more transferable mathematical reasoning patterns than reward maximization.

Importantly, while PRM training has a computational cost, our analysis, detailed in subsection H.4, shows that GFlowNet fine-tuning itself is surprisingly efficient, requiring less training time than comparable PPO baselines.

To further demonstrate the effectiveness of our data augmentation technique, we conducted an additional fine-tuning experiment, as shown in Table 3. In this setup, we fine-tune Llama3.2-3B-it with an open-source PRM, namely Skywork-o1-Open-PRM-Qwen-2.5-7B (He et al., 2024), and compare the outcomes with those obtained using our own PRM.

Table 3: Performance comparison on mathematical reasoning benchmarks between our PRM and Skywork-o1-Open-PRM-Qwen-2.5-7B

| Model | SAT MATH | GSM8K |
|---|---|---|
| Llama3.2-3B-it + GFlowNet | 65.6% | 67.8% |
| + Our finetuned Qwen2.5-7B-math | **75.0%** | 68.5% |
| + Open-PRM-Qwen-2.5-7B | 68.8% | 68.5% |

## 5.2 SOLUTION DIVERSITY ANALYSIS

To quantitatively evaluate the diversity of solutions generated by GFlowNet fine-tuning, we used a semantic similarity metric. We chose semantic embeddings over lexical measures for mathematical reasoning diversity assessment because: (1) lexical metrics like Levenshtein distance or Jaccard similarity capture only surface-level textual differences, missing semantic equivalence in mathemat-

ical expressions (e.g., "$x = 4$" vs "$x = 2^2$"), and (2) mathematical reasoning diversity requires understanding conceptual differences between solution approaches, not just word-level variations.

We measured the pairwise semantic similarity between the reasoning steps generated by each model using the "paraphrase-MiniLM-L6-v2" model (Reimers & Gurevych, 2019), a pre-trained sentence embedding model specifically designed to capture semantic relationships. Each step was encoded into a high-dimensional space using the encoding function of the model, with the output stored as tensors. The pairwise similarity between these embeddings was then computed using the cosine similarity measure, a standard metric for assessing vector similarity in semantic spaces. This metric provides a continuous score representing the semantic proximity between different reasoning steps, where lower scores indicate greater dissimilarity and, consequently, higher solution diversity. The average semantic similarity across multiple generated solutions for each model is presented in Table 4, where lower scores indicate greater diversity, and results averaged across 1,000 problems from MATH.

The semantic similarity scores presented in Table 4 provide compelling evidence that GFlowNet fine-tuning effectively enhances the diversity of generated mathematical solutions. This consistent reduction in semantic similarity indicates that GFlowNet-fine-tuned models are indeed generating reasoning paths that are more semantically distinct and varied compared to the other approaches. This result is consistent with the observations made during the finetuning of the LLM with

Table 4: Solution diversity analysis.

| Model | Avg. Semantic Similarity |
|---|---|
| Llama3.2-3B-it | 0.80 |
| + PPO | 0.82 |
| + GFlowNet | **0.78** |

GFlowNets, as shown in Figure 4c. Specifically, we see that during training, the LLM progressively generates steps whose probabilities become increasingly aligned with the rewards provided by the PRM (i.e., the probabilities assigned by the LLM to the tokens of a step get closer to the PRM rewards for that step). Consequently, this enhanced diversity is a direct consequence of the GFlowNet training objective, which, unlike reward-maximizing RL methods like PPO, is explicitly designed to sample from a distribution proportional to the reward. A concrete example of this induced diversity is provided in Appendix I.

## 6 CONCLUSION

We have introduced a novel step-level GFlowNet framework for mathematical reasoning that achieves two crucial objectives: improving accuracy and promoting solution diversity. Our approach demonstrates that operating at the reasoning step level, rather than the token level, enables more effective control over the generation process while maintaining semantic coherence.

The empirical results show significant improvements over both baseline models and PPO fine-tuning, particularly for smaller models. This suggests that our approach could be especially valuable in resource-constrained settings where smaller models are preferred. Furthermore, the increased solution diversity achieved by our method aligns well with educational applications, where exposure to multiple valid solution strategies can enhance learning outcomes.

**Limitations.** Despite these promising results, several limitations warrant discussion. The computational cost of initial PRM training through MCTS remains significant, though this is a one-time expense that enables subsequent efficient fine-tuning. The approach requires careful tuning of replay buffer strategies and similarity threshold parameters. Additionally, while our similarity-based augmentation partially addresses potential biases in automated PRM training, more sophisticated bias detection and mitigation strategies remain important areas for development.

**Future Work.** Promising directions include exploring offline GFlowNet training to reduce computational costs, developing more sophisticated semantic diversity metrics tailored for mathematical reasoning, investigating the educational impact of diverse solution generation, and extending the approach to other complex reasoning domains such as program synthesis and theorem proving.

REPRODUCIBILITY STATEMENT

To ensure full reproducibility of our results, we provide comprehensive implementation details and make all necessary resources publicly available. The complete source code for our methodology, including PRM training via MCTS data generation, similarity-based data augmentation, and step-level GFlowNet fine-tuning, is available at https://anonymous.4open.science/r/gfn-F329. Our implementation includes four main components: (1) seed dataset generation scripts for initial data collection, (2) MCTS-based training data generation as detailed in Algorithm 1, (3) similarity-based data augmentation procedures described in Section 3.2, and (4) GFlowNet fine-tuning implementation with SubTB loss as presented in Algorithm 2. All hyperparameters, including learning rates, batch sizes and Replay Buffer Size are explicitly documented in Section G.1 and provided as default values in our code. The data processing pipeline, from initial seed generation through final PRM training dataset creation, is fully automated and includes detailed logging for verification. Additionally, we provide evaluation scripts for PRM-guided search validation and comprehensive testing procedures that allow researchers to reproduce all experimental results reported in Section 5. The computational requirements, hardware specifications, and expected training times are documented in Section H.4 to facilitate resource planning for reproduction studies.

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

## A    LLM USAGE

We used LLMs to aid and polish writing throughout this manuscript. Specifically, we employed large language models for grammatical refinement, clarity improvements, and ensuring consistency in technical terminology across sections. All scientific content, experimental design, results, and interpretations represent original work by the authors, with LLMs serving solely as writing assistants for language enhancement.

## B    BROADER IMPACT AND APPLICATIONS

Our diversity-aware approach has significant implications beyond mathematical reasoning. In educational contexts, systems using our method could better evaluate student work that differs from canonical solutions but remains mathematically sound. The step-level GFlowNet principles could generalize to other sequential reasoning domains, such as coding tasks where lines of code could be treated analogously to reasoning steps, or logical proofs where intermediate deductions follow similar hierarchical structures.

## C    EXTENDED RELATED WORK

**Mathematical Reasoning in LLMs.**    Recent 2024 developments include multimodal approaches like MultiMath (Peng et al., 2024), and advanced training methods like Flow of Reasoning (Yu et al., 2025), which enables diverse solution generation with minimal training examples. However, critical studies like GSM-Symbolic (Mirzadeh et al., 2025) reveal fundamental limitations in current LLMs, showing performance degradation when numerical values or problem structure vary, indicating reliance on pattern matching rather than genuine reasoning.

**Generative Flow Networks.**    GFlowNets have been successfully applied to molecule generation (Jain et al., 2022), Bayesian structure learning (Deleu et al., 2022), causal discovery (Manta et al., 2023; Deleu et al., 2022), material design (Cipcigan et al., 2024; Nguyen et al., 2023), drug discovery (Pandey et al., 2024), biological sequence design and editing (Jain et al., 2022; Ghari et al., 2024), scheduling (Zhang et al., 2023a), and combinatorial optimization problems (Zhang et al., 2023b). Recent theoretical extensions have expanded GFlowNets to continuous spaces (Lahlou et al., 2023), stochastic environments (Pan et al., 2023), and adversarial settings (Jiralerspong et al., 2024).

## D    RELATIONSHIP TO MAXIMUM ENTROPY RL

Recent theoretical work (Tiapkin et al., 2024) establishes that GFlowNets are equivalent to maximum entropy RL when the underlying DAG forms a tree structure. In sequential mathematical reasoning, this raises the question of why our approach outperforms standard PPO.

We note that the theoretical equivalence holds for **maximum entropy RL** (where entropy maximization is the primary objective), not standard PPO with entropy regularization. While our PPO baseline includes entropy regularization (coefficient 0.01), this serves primarily as an exploration bonus rather than implementing the full maximum entropy framework where entropy is co-optimized with reward.

The key differences are: (1) **Entropy Treatment**: GFlowNets naturally sample proportionally to rewards (maximum entropy principle), while our PPO uses entropy as a small regularization term; (2) **Training Objective**: GFlowNets directly optimize for reward-proportional sampling, while PPO maximizes expected return with entropy bonus; (3) **Exploration Dynamics**: The SubTB loss provides different exploration dynamics than clipped PPO objectives.

This distinction explains our empirical improvements: even when theoretically related, the implementation details and training dynamics create meaningful practical differences.

# E  PROCESS REWARD MODEL TRAINING

## E.1  MCTS DATA GENERATION PROCESS

Our MCTS-based data generation extends beyond the binary evaluation framework of Luo et al. (2024) to generate rich step-level training data with continuous quality scores. This section details the technical implementation and algorithmic components.

### E.1.1  STEP IDENTIFICATION AND PREPROCESSING

Mathematical reasoning solutions are decomposed into individual steps using line breaks as delimiters. The data generation model (Qwen2.5-Math) is instructed through 4-shot prompting to structure its reasoning with clear step separations, ensuring consistent parsing across the dataset.

### E.1.2  MONTE CARLO EVALUATION PROTOCOL

The core evaluation mechanism assesses each step $s$ through multiple completion attempts. Starting from the solution prefix that includes all steps up to $s$, we generate $k = 96$ independent rollouts using temperature sampling (T=0.6) to promote diversity. Each rollout produces a complete solution attempt, and success is determined by final answer correctness:

$$MC(s) = \frac{\text{successful rollouts from prefix ending at step s}}{k}$$

This continuous scoring scheme provides granular quality assessment, distinguishing between steps that consistently lead to success (MC = 1.0), those with moderate reliability (MC $\in (0, 1)$), and unreliable steps (MC = 0.0).

### E.1.3  TREE SEARCH AND ROLLOUT MANAGEMENT

Generated rollouts are organized within an MCTS tree structure, with each node maintaining three key statistics:

1. $N(s)$: Visit count for state $s$
2. $MC(s)$: Monte Carlo evaluation score
3. $Q(s, r)$: State-rollout value function incorporating both quality and length considerations:

$$Q(s,r) = \alpha^{1-MC(s)}\beta^{\frac{\text{len(r)}}{L}}$$

where hyperparameters $\alpha, \beta \in (0, 1]$ balance quality preference against rollout length, and $L > 0$ normalizes length effects.

### E.1.4  SELECTION STRATEGY AND EXPLORATION

Rollout selection follows an adapted PUCT mechanism that balances exploitation of high-quality paths with exploration of under-visited regions:

$$(s, r) = \arg\max_{(s,r)}[Q(s,r) + U(s)]$$

The exploration term $U(s)$ encourages diverse tree construction:

$$U(s) = c_{puct}\frac{\sqrt{\sum_i N(s_i)}}{1 + N(s)}$$

where $c_{puct}$ controls exploration intensity. This strategy initially favors rollouts with low visit counts but gradually shifts preference towards those with high rollout values. By prioritizing high-quality reasoning steps (higher $MC(s)$ values), we create more effective training data for the PRM compared to uniform sampling of potentially poor-quality steps.

### E.1.5 DATASET CONSTRUCTION AND TERMINATION

The algorithm 1 constructs training examples by exploring reasoning trees until error detection through binary search. When a step receives $MC(s) = 0$ (no successful rollouts), it marks the first reasoning error, and tree exploration for that path terminates. All valid steps preceding this error point are included in the final training dataset with their computed MC scores.

The complete process terminates when the rollout pool is exhausted or computational limits are reached, yielding a comprehensive dataset of step-quality pairs for PRM training.

---

**Algorithm 1:** MCTS-based data generation

---

**Require:** Question $q$, Language Model LM, Number of completions $k = 96$, Temperature $T = 0.6$
**Ensure:** Dataset of training examples
1: Initialize root state $r_{\text{root}} \leftarrow q$
2: Initialize tree with root node $s_{\text{root}}$ containing $r_{\text{root}}$
3: Initialize visit count $N(s_{\text{root}}) \leftarrow 0$
4: Initialize Monte Carlo estimation $MC(s_{\text{root}}) \leftarrow 0$
5: Initialize state-rollout value function $Q(s_{\text{root}}, r) \leftarrow 0$
6: **while** not converged **do**
7:     Select a trajectory using PUCT algorithm based on $Q(s, r)$ and $U(s)$
8:     Perform binary search to locate the first incorrect step in the selected trajectory, the step currently sampled is $s_{\text{candidate}}$
9:     Generate $k$ completions using temperature sampling from $\text{LM}(c|s_{\text{candidate}})$
10:     Initialize $correct\_count \leftarrow 0$
11:     **for** each completion $c_i$ **do**
12:       **if** rollout is correct **then**
13:         $correct\_count \leftarrow correct\_count + 1$
14:       **end if**
15:     **end for**
16:     $MC(s_{\text{candidate}}) \leftarrow \frac{correct\_count}{k}$
17:     **if** $correct\_count = 0$ **then**
18:       Add $s_{\text{candidate}}$ to the dataset with value 0
19:       This step is considered to be incorrect
20:       Break
21:     **else**
22:       Update tree by adding $s_{\text{candidate}}$ with value $MC(s_{\text{candidate}})$ and add all the generated rollouts to the tree as well as the number of visits of the node $N(s_{\text{candidate}})$ and $Q(s_{\text{candidate}}, r)$
23:     **end if**
24: **end while**
25: Dataset Collection Phase:
26: **for** each node $s$ in the tree **do**
27:     Add $s$ to the dataset with its evaluated step, value and all the rollouts generated from this step.
28: **end for**
**Ensure:** Dataset of training examples

---

### E.2 HYPERPARAMETERS

Training was conducted for 1 epoch with a learning rate of 8e-6, batch size 144, and weight decay 0.01 on 8 A100 GPUs.

### E.3 TRAINING DATASET SAMPLES

Table 5 illustrates representative examples from our PRM training dataset, showcasing the diversity of mathematical reasoning problems and the range of step quality scores generated through our MCTS-based approach and similarity-based augmentation.

Table 5: Sample entries from our PRM training dataset generated via MCTS and similarity-based augmentation. Each entry contains a mathematical problem, partial solution context, candidate next step, and ground truth step quality score derived from Monte Carlo evaluations.

| Question | Solution Context | Candidate Step | MC Score |
|---|---|---|---|
| The product of two whole numbers $a$ and $b$ is equal to 8 times their sum. If $a$ is 3 more than $b$, what is the value of $a$? | We are given that $ab = 8(a + b)$ and $a = b + 3$. Substituting the second equation into the first, we get $(b + 3)b = 8(b + 3 + b)$. Simplifying, we get $b^2 + 3b = 8(2b + 3)$. Expanding, we get $b^2 + 3b = 16b + 24$. Rearranging, we get $b^2 - 13b - 24 = 0$. Factoring, we get $(b - 16)(b + 1) = 0$. Therefore, $b = 16$ or $b = -1$. Since $b$ is a whole number, $b = 16$. Then $a = b + 3 = 16 + 3 = 19$. | So the value of $a$ is $\boxed{19}$. | 1.00 |
| In a certain country, the postal system uses a unique three-letter code for each city. The code consists of three different letters, and the letters are chosen from a set of $n$ letters, where $n$ is a positive integer. Given that there are exactly 120 different codes that can be formed using these $n$ letters, find the value of $n$. | The number of ways to choose the first letter is $n$. The number of ways to choose the second letter is $n - 1$, since we cannot repeat the first letter. The number of ways to choose the third letter is $n - 2$, since we cannot repeat the first two letters. | Therefore, the total number of different codes is $n(n - 1)(n - 2)$. | 0.89 |
| A triangle $ABC$ has vertices $A(1, -2)$, $B(9, -2)$, and $C(x, y)$. The point $C$ lies on the parabola $x = y^2 + 1$. Find the minimum possible perimeter of triangle $ABC$. | The distance between $A$ and $B$ is $AB = 8$. The distance between $A$ and $C$ is $AC = \sqrt{(x - 1)^2 + (y + 2)^2}$ and the distance between $B$ and $C$ is $BC = \sqrt{(x - 9)^2 + (y + 2)^2}$. The perimeter of triangle $ABC$ is $P = AB + AC + BC = 8 + \sqrt{(x - 1)^2 + (y + 2)^2} + \sqrt{(x - 9)^2 + (y + 2)^2}$. Since $C$ lies on the parabola $x = y^2 + 1$, we can substitute $x = y^2 + 1$ into the expression for $P$ to get $P = 8 + \sqrt{y^4 + 4y^2 + 4} + \sqrt{y^4 - 16y^2 + 64}$. We can simplify this expression by noticing that $y^4 + 4y^2 + 4 = (y^2 + 2)^2$ and $y^4 - 16y^2 + 64 = (y^2 - 8)^2$. | Therefore, the perimeter is $P = 8 + \sqrt{(y^2 + 2)^2} + \sqrt{(y^2 - 8)^2} = 8 + y^2 + 2 + y^2 - 8 = 2y^2$. | 0.00 |

*Note: Monte Carlo (MC) scores represent ground truth labels derived from MCTS rollout evaluations, ranging from 0.00 (steps leading to incorrect solutions) to 1.00 (steps leading to correct solutions). These scores serve as training targets for the PRM. The similarity-based augmentation process ensures consistent labeling across semantically equivalent reasoning steps.*

### E.4 ERROR DETECTION VALIDATION

We systematically corrupted correct solution steps in two ways and evaluated PRM responses:

**Number Manipulation**: For the step "Calculate: $24 - 6 = 18$", corrupting to "$24 - 6 = 16$" reduced PRM score from 0.84 to 0.05, demonstrating sensitivity to arithmetic errors.

**Logic Corruption**: For the step "Since $x^2 = 16$, we have $x = 4$ or $x = -4$", removing the negative solution ("$x = -4$") reduced PRM score from 0.89 to 0.34, showing detection of incomplete logical reasoning.

### E.5 PRM Multimodality Validation

To validate our PRM's capability to recognize multiple valid reasoning approaches, we evaluated its performance on problems with diverse solution strategies. Our trained PRM assigns comparable high scores to different valid reasoning steps while correctly identifying and penalizing incorrect steps with low scores, as illustrated in Figure 3.

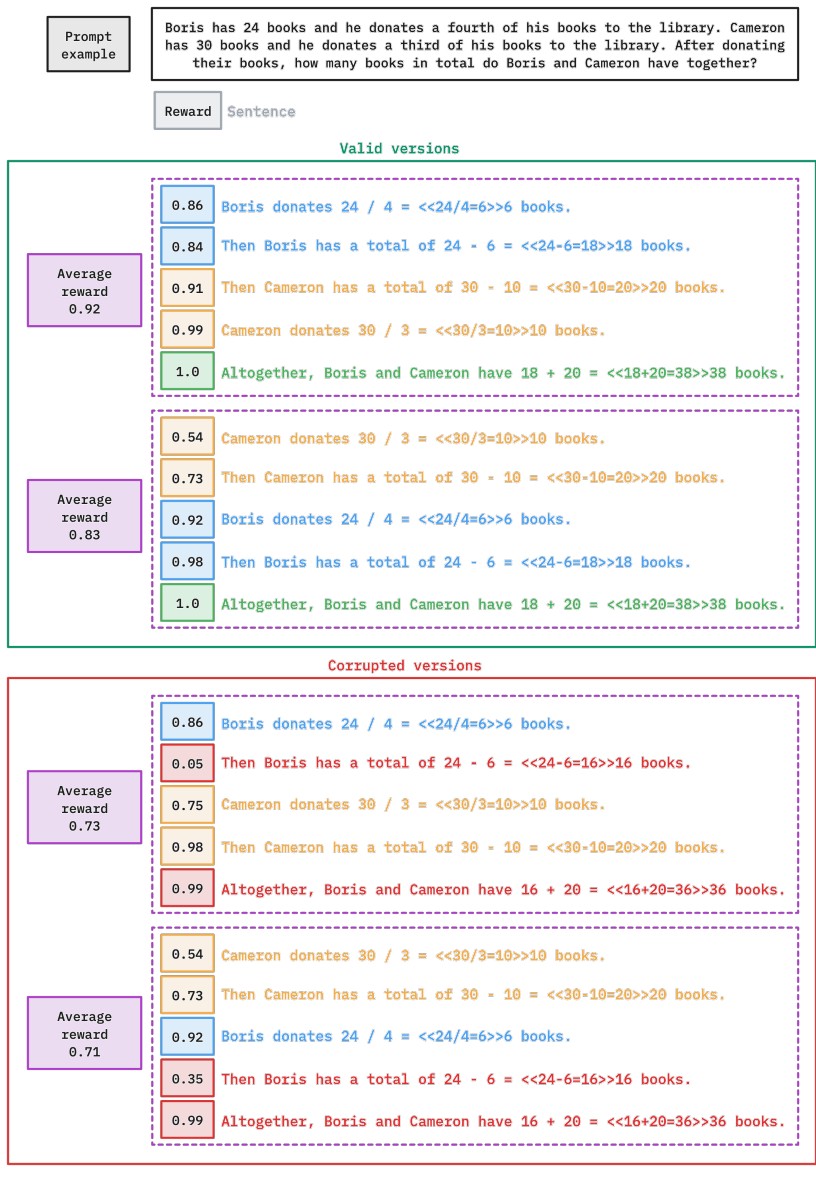

Figure 3: Reasoning steps and corresponding PRM scores. Valid steps from different approaches receive high, comparable scores, while corrupted steps receive lower scores.

### E.6 PRM Limitations and Bias Considerations

While our automated PRM training eliminates human annotation costs, it may introduce certain biases. The MCTS-based data generation could favor certain reasoning patterns over others, and the similarity-based augmentation might propagate systematic errors. To partially mitigate these concerns, our similarity grouping approach creates more diverse training examples and helps identify inconsistencies in step valuations. However, we acknowledge that more sophisticated bias detection and mitigation strategies, such as consensus filtering or adversarial validation, remain important areas for future work.

## F Bayesian Posterior Sampling Framework

Our approach can be viewed as performing Bayesian posterior sampling over reasoning paths, following the framework established by Deleu et al. (2022) for Bayesian structure learning. In this interpretation, we seek to sample diverse reasoning trajectories from a posterior distribution that combines a prior over reasoning paths with evidence from our PRM.

Formally, let $\tau = (s_0, s_1, ..., s_n)$ represent a reasoning trajectory, where $s_0$ is the initial question and $s_n$ is the complete solution. We define:

- **Prior**: $P(\tau)$ represents the pretrained LLM's distribution over reasoning paths, encoding learned mathematical reasoning patterns
- **Likelihood**: $L(\tau|PRM) = \prod_{i=1}^{n} U(s_i|s_{i-1})$ represents the evidence from our PRM about each reasoning step
- **Posterior**: $P(\tau|PRM) \propto P(\tau) \cdot L(\tau|PRM)$ combines pretrained knowledge with PRM evidence

**Justification for PRM as Likelihood**: The PRM scores $U(s_i|s_{i-1})$ naturally function as likelihood terms because they represent the probability of observing a "correct" reasoning step given the context. In Bayesian terms, we can interpret this as $P(\text{step is correct}|s_i, s_{i-1}, PRM)$. Since our PRM is trained to predict the probability that a step leads to a correct solution (using MCTS-derived ground truth), these scores directly quantify the evidence that each step provides toward the trajectory being correct. The multiplicative structure $\prod_{i=1}^{n} U(s_i|s_{i-1})$ reflects the conditional independence assumption that step correctness depends primarily on local context, which is reasonable for mathematical reasoning where each step builds incrementally on previous work.

The GFlowNet framework naturally implements this Bayesian updating by learning to sample trajectories with probabilities proportional to the reward $R(\tau) = P(\tau) \cdot L(\tau|PRM)$. This formulation provides a principled foundation for diverse reasoning: rather than seeking a single optimal solution (as in traditional RL), we sample from the full posterior distribution, naturally capturing multiple high-quality reasoning strategies.

## G GFlowNet Training Implementation

### G.1 Experimental Configuration

We conduct experiments on Llama3 (Dubey et al., 2024) using two model sizes:

- Llama3.2-3B-it: A smaller model to demonstrate efficiency
- Llama3.1-8B-it: A medium-sized model for performance comparison

**Hyperparameters:**

- **Learning Rate**: We used a learning rate of 5e-6 for the Adam optimizer during GFlowNet fine-tuning. To optimize the learning rate schedule, we used a cosine scheduler, which gradually decreases the learning rate over the course of the training epoch.
- **Batch Size**: A batch size of 144 trajectories was used for each training iteration. This batch size was chosen to balance computational efficiency and the stability of gradient updates.

- **Gradient Clipping**: To prevent exploding gradients during training, we applied gradient clipping with a maximum norm of 1.0. This technique helps to stabilize the training process, particularly in the context of recurrent neural networks like the LLMs used in our GFlowNet policy.

- $\lambda$ **Value for SubTB Loss**: The $\lambda$ hyperparameter in the Subtrajectory Balance (SubTB) loss function controls the discount factor for subtrajectory rewards. We set $\lambda = 1.0$ in our experiments. This value implies no discounting, giving equal weight to all subtrajectory balance terms in the loss.

- **Replay Buffer Size**: To stabilize training and improve sample efficiency, we utilized a replay buffer of size 1000. This buffer stores previously generated complete trajectories, allowing the GFlowNet policy to learn from a diverse set of high-reward experiences.

### G.2 STEP-LEVEL GFLOWNET ALGORITHM

Our step-level GFlowNet fine-tuning procedure leverages the trained PRM for reward evaluation and employs a prioritized replay buffer to enhance training efficiency and solution diversity.

---

**Algorithm 2:** GFlowNet Fine-tuning

---

**Require:** Question $q$, Policy Model $\pi_\theta$ (LLM), PRM $U$, Replay Buffer $\mathcal{B}$, Generations per question $k$, Temperature $T$, Batch Size $B$

1: **for** each batch of questions in the training dataset **do**
2:     **for** each question $q$ in the batch **do**
3:         Generate $k$ responses for question $q$ using $\pi_\theta$ with temperature $T$
4:         Split each response into steps and evaluate reward $R(s_{1:i}s_f)$ for each trajectory using the PRM, storing trajectories in $\mathcal{B}$
5:         Sample a batch of size $B$ from the replay buffer $\mathcal{B}$
6:         **for** each trajectory in the batch **do**
7:             Compute the loss $\mathcal{L}$ based on the reward function $R$ and policy model $\pi_\theta$
8:         **end for**
9:         Perform one step of optimization to minimize the loss $\mathcal{L}$ with respect to the parameters $\theta$ of $\pi_\theta$
10:     **end for**
11: **end for**

**Ensure:** Trained policy model $\pi_\theta$ with GFlowNets

---

### G.3 TRAINING DYNAMICS AND OPTIMIZATION

Our step-level GFlowNet training incorporates several key optimizations to improve convergence and sample efficiency:

**Prioritized Replay Buffer**: Instead of uniform sampling, we implement a prioritized replay mechanism that favors trajectories with higher PRM-evaluated rewards. This prioritization accelerates learning by focusing on high-quality reasoning experiences, similar to prioritized experience replay in deep RL but adapted for our reward-proportional sampling objective.

**Temperature Scheduling**: We use temperature sampling ($T = 0.6$) during trajectory generation to balance exploration and exploitation. This temperature is carefully tuned: higher values ($T > 0.8$) lead to excessive exploration and incoherent steps, while lower values ($T < 0.4$) result in mode collapse, reducing solution diversity.

**Gradient Stabilization**: The SubTB loss can exhibit high variance due to the multiplicative reward structure. We apply gradient clipping (max norm 1.0) and use a learning rate schedule that reduces volatility while maintaining effective parameter updates. The discount factor $\lambda = 1.0$ gives equal weight to all subtrajectories, ensuring comprehensive coverage of the reasoning space.

## H  TRAINING DYNAMICS ANALYSIS

This section presents a comprehensive analysis of the training dynamics during GFlowNet fine-tuning of our language model. We monitor three key metrics throughout the training process to assess convergence behavior and validate the effectiveness of our Sub-TB learning approach.

### H.1  MONITORED METRICS

We track the following metrics during training to provide insights into the learning dynamics:

- **Sub-TB Loss**: The Sub-Trajectory Balance (Sub-TB) loss function used to train the GFlowNet policy. This loss ensures that the flow conservation constraint is satisfied at each step of the trajectory, enabling the model to learn proper sampling probabilities proportional to rewards.
- **Average Reward**: The mean reward signal obtained from the PRM across sampled trajectories during training. This metric reflects the quality of reasoning steps generated by the model.
- **Proportionality Gap**: A critical alignment metric measuring the absolute deviation between token selection probabilities and their corresponding PRM rewards.

### H.2  PROPORTIONALITY GAP: A KEY ALIGNMENT METRIC

The proportionality gap directly measures how well the model achieves the fundamental GFlowNet objective of sampling trajectories with probabilities proportional to their rewards. Following our implementation, this metric is computed as:

$$\text{Proportionality Gap} = \mathbb{E}_{t \sim \tau} \left[ |p_t - r_t| \right] \tag{2}$$

where $p_t$ represents the probability assigned by the model to selecting the token at step $t$, and $r_t$ is the corresponding reward from the PRM for that step.

This metric is fundamental to validating GFlowNet training effectiveness. The core principle of GFlowNets requires that sampling probabilities should be proportional to rewards. Therefore, a decreasing proportionality gap indicates successful learning of this alignment. When this gap approaches zero, it signifies that the model has learned to assign higher probabilities to reasoning steps that receive higher rewards from the PRM, which is precisely the desired behavior for effective mathematical reasoning.

The reduction of this gap throughout training provides direct evidence that our Sub-TB loss successfully guides the model toward the optimal sampling distribution, where high-quality reasoning paths are preferentially explored.

### H.3  TRAINING PROGRESSION ANALYSIS

Figure 4 illustrates the evolution of these three critical metrics throughout the GFlowNet fine-tuning process. The training dynamics reveal several important characteristics that validate our approach.

### H.4  COMPUTATIONAL EFFICIENCY ANALYSIS

To assess the computational efficiency of our GFlowNet fine-tuning approach, we conducted a comparative analysis of the training time and computational resources required for PRM training, GFlowNet fine-tuning, and PPO baseline training. These experiments were performed using consistent hardware and training data settings to ensure a fair comparison. The results of this training efficiency comparison are summarized below:

**Computational Resources:** All training experiments were conducted on a cluster of machines equipped with NVIDIA A100 GPUs. For PRM training, we utilized 8 A100 GPUs in parallel to accelerate the data generation and model fine-tuning process. GFlowNet and PPO fine-tuning experiments were conducted using the same hardware setup for consistent resource allocation.

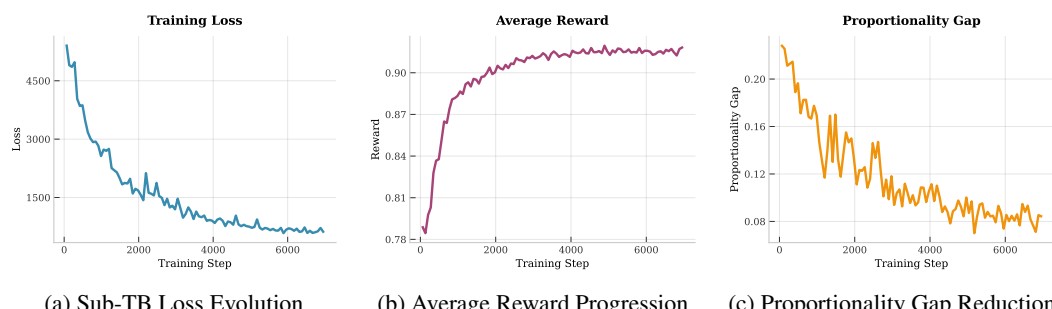

| (a) Sub-TB Loss Evolution | (b) Average Reward Progression | (c) Proportionality Gap Reduction |

Figure 4: Training dynamics during GFlowNet fine-tuning of Llama3.2-3B-it showing (a) Sub-TB loss convergence, (b) average reward improvement, and (c) proportionality gap reduction. The consistent decrease in proportionality gap demonstrates successful alignment between token selection probabilities and PRM rewards, validating the effectiveness of our Sub-TB learning approach.

**Training Time Comparison:**

- **PRM Training**: Training the Process Reward Model (PRM), including the automated data generation phase using MCTS and the subsequent PRM fine-tuning, required approximately 4 hours of training time using 8 A100 GPUs. The data generation phase using MCTS constitutes a significant portion of this training time.

- **GFlowNet Fine-tuning**: Fine-tuning a GFlowNet policy for a specific LLM (e.g., Llama3.2-3B-it or Llama3.1-8B-it) using our step-level approach typically required around 1 hour of training time on the allocated hardware for 10,000 questions from the OpenMath-Instruct2 dataset (Toshniwal et al., 2024).

- **PPO Baseline Training**: Training the PPO baseline models, using the same PRM for reward guidance and with comparable hyperparameter settings, generally required approximately 2 hours of training time for 10,000 questions from the OpenMathInstruct2 dataset (Toshniwal et al., 2024). This is longer than the GFlowNet fine-tuning time, potentially indicating a greater sample efficiency or faster convergence of the GFlowNet training process in our setup.

## I  DIVERSITY EXAMPLE

To illustrate the diversity induced by GFlowNet fine-tuning concretely, we evaluate both approaches under identical generation conditions. Using consistent sampling parameters, consider solving "Find the value of $x$ if $2x + 3 = 11$":

**PPO Solutions** (consistent trajectory, lexical variations):

- "Subtract 3 from both sides: $2x = 8$. Divide by 2: $x = 4$."
- "First subtract 3 from each side: $2x = 8$. Then divide both sides by 2: $x = 4$."

**GFlowNet Solutions** (distinct valid trajectories):

- "Subtract 3 from both sides: $2x = 8$. Divide by 2: $x = 4$."
- "Divide the entire equation by 2: $x + 1.5 = 5.5$. Subtract 1.5: $x = 4$."

While PPO converges to a single approach with minor linguistic variations, GFlowNet generates genuinely distinct mathematical pathways, demonstrating exploration of diverse solution strategies rather than exploitation of a single optimal path.

