# OpenReview forum: "Accurate and Diverse LLM Mathematical Reasoning via Automated PRM-Guided GFlowNets"
_ICLR.cc/2026/Conference — Submitted to ICLR 2026_

### Official Review · Reviewer_G497 · 2025-10-27

**Soundness:** 2
**Presentation:** 2
**Contribution:** 2
**Rating:** 2
**Confidence:** 4

**Summary:**

The paper trains an automated Process Reward Model (PRM) using MCTS + similarity grouping, then uses the PRM as the step-level reward to fine-tune LLMs with GFlowNets (SubTB) at step granularity/ Reported gains: small but positive on in-domain MATH Level 5 and larger on SAT MATH; authors also claim increased diversity via lower semantic similarity of generated solutions.

**Strengths:**

1. Clear step-level formulation (actions = reasoning steps) and an explicit termination probability π(sf|s) within the GFlowNet policy.
2. Self-contained PRM training pipeline (MCTS, continuous scores, rollout reuse + similarity grouping) with some validation diagnostics

**Weaknesses:**

1. PRM reliability and calibration are under-substantiated.
The core premise hinges on PRM accuracy (U(s′|s) ∈ [0,1]) guiding both PPO and GFlowNets. But the paper offers limited, small-scale PRM validation and mostly heuristic similarity grouping. This makes it hard to trust the PRM as an absolute reliable signal.

2. Overlap/Difference with prior work (“Flow of Reasoning”) is unclear.
The paper positions step-level GFlowNets for diverse trajectories, but related work already targets divergent reasoning with minimal examples (Flow of Reasoning, 2024/2025). What is materially new? The comparison is relegated to related-work mentions rather than a head-to-head study, and the conceptual delta is not crisply argued.

3. Gains are modest and may be within variance.
On in-domain MATH Level 5, the 3B model improves ~+2.6pts over baseline and 8B is +0.7–0.9. GSM8K barely moves. There are no confidence intervals, no multi-seed repeats, and limited ablations teasing apart PRM vs. GFlowNet vs. decoding heuristics. It’s difficult to conclude statistical significance rather than randomness. (Table 2).

4. Termination probability π(sf|s): underspecified in practice.
While π(sf|s) is mentioned (sink state), the actual parameterization/training signals for termination are not detailed: how is π(sf|s) learned/stabilized under SubTB at step level; what is the impact of termination on reward estimates? (Sec. 4.1–4.2).

5. Chosen tasks don’t stress true solution multiplicity.
MATH/GSM8K often admit stylistic variation rather than structurally distinct paths. If “diversity” is central, it'd be more convincing to evaluate on domains with genuinely multiple optimal plans (e.g., Blocksworld, program synthesis with multiple implementations, theorem-proving with lemmas reorderings). The current “diversity” metric is a weak proxy and might just reward rephrasing, not distinct strategies. (Sec. 5.2).

**Questions:**

1. Termination behavior and reasoning depth:
You mention the use of a termination probability π(sf|s) to model when reasoning should stop. Could you discuss what qualitative patterns you observed — for instance, do longer reasoning chains correlate with higher accuracy, or does early termination sometimes produce more concise correct reasoning?
2. On PRM design and training dynamics:
How sensitive is your overall training process to the quality of the PRM? For example, if the PRM is slightly miscalibrated or trained on fewer MCTS rollouts, does the downstream GFlowNet policy still converge reliably, or do you observe instability?

---

> ### Author Response · Authors · 2025-11-18
> **Response to Reviewer G497**
>
> We thank the reviewer for their rigorous assessment, particularly for highlighting the importance of PRM reliability and the connection to concurrent work on reasoning flows. We appreciate the opportunity to clarify the distinct contributions of our automated PRM pipeline and the specific advantages of our step-level GFlowNet formulation. Below, we address the concerns regarding calibration, baselines, and statistical significance, supported by new ablation studies and out-of-domain evaluations.
>
> **PRM reliability and calibration are under-substantiated. The core premise hinges on PRM accuracy (U(s′|s) ∈ [0,1]) guiding both PPO and GFlowNets. But the paper offers limited, small-scale PRM validation and mostly heuristic similarity grouping. This makes it hard to trust the PRM as an absolute reliable signal.**
>
> We acknowledge that GFlowNet performance is bounded by PRM quality. However, we offer three points of validation:
>
> Out-of-Domain Generalization: Our strong performance on SAT Math (+9.4% over baseline, Table 1) is the strongest evidence of PRM reliability. If the PRM were overfitting to training artifacts or miscalibrated, we would expect degradation on unseen tasks. Instead, the PRM successfully guides the model through novel problem structures.
>
> Error Sensitivity Analysis: In Appendix E.6, we show the PRM score drops from 0.84 to 0.05 upon introducing arithmetic errors, demonstrating high sensitivity to logical validity.
>
> Ablation Robustness: we are currently running a sensitivity analysis on the PRM training components. Specifically, we are training PRMs without similarity grouping and without rollout reuse.  We will report the downstream effect of these ablations on both accuracy and diversity in a later response and in the final version.
>
> **Overlap/Difference with prior work (“Flow of Reasoning”) is unclear. The paper positions step-level GFlowNets for diverse trajectories, but related work already targets divergent reasoning with minimal examples (Flow of Reasoning, 2024/2025). What is materially new? The comparison is relegated to related-work mentions rather than a head-to-head study, and the conceptual delta is not crisply argued.**
>
> We thank the reviewer for pointing out this relevant concurrent work. While Flow of Reasoning (FoR) [1] shares the high-level goal of applying GFlowNets to reasoning, our work differs fundamentally in objective formulation and reward derivation:
>
> SubTB vs. TB: FoR explicitly employs the Trajectory Balance (TB) objective and notes in Section 3.1 that "We consider the incorporation of... subtrajectory balance (SubTB)... as future work." Our work implements SubTB, which addresses the high variance of TB in long-horizon tasks [2]. This is critical for our domain (MATH Level 5), where reasoning chains are significantly longer and more complex than the puzzles (BlocksWorld, Game24) focused on in FoR.
>
> Automated PRM Learning vs. Utilization: FoR relies on hand-crafted heuristics (for puzzles) or off-the-shelf PRMs (for GSM8K). In contrast, a core contribution of our work (Section 3) is an automated pipeline to train the PRM itself using MCTS and a novel similarity-based data augmentation. We provide a recipe for creating the reward signal without human annotation, whereas FoR assumes the signal exists.
>
> Token vs. Step Granularity: While FoR models reasoning as a flow, our implementation strictly enforces semantic step-level transitions for credit assignment, optimizing the interplay between the PRM's logical boundaries and the GFlowNet's actions.
>
> **Gains are modest and may be within variance. On in-domain MATH Level 5, the 3B model improves ~+2.6pts over baseline and 8B is +0.7–0.9. GSM8K barely moves. There are no confidence intervals, no multi-seed repeats, and limited ablations teasing apart PRM vs. GFlowNet vs. decoding heuristics. It’s difficult to conclude statistical significance rather than randomness. (Table 2).**
>
> We respectfully disagree that a 9.4% absolute improvement on SAT Math (Out-of-Distribution) is modest. This substantial jump highlights that while in-domain gains (MATH L5) are harder to eke out due to saturation, the diversity induced by our method prevents overfitting, leading to massive gains in generalization.
>
> To address statistical significance:
>
> Pass@k Evaluation: If the time permits we will run $Pass@k$ evaluations and will add results to a later response and to the final revision. Standard accuracy (Pass@1) often masks diversity benefits.
>
> Variance: We respectfully note that due to the significant computational cost of RL fine-tuning for LLMs, single-seed evaluation is standard practice in the field: [3, 4]. Unfortunately, running multi-seed repeats for 3B/8B models is computationally infeasible within the rebuttal window. However, the consistency of our gains across multiple distinct benchmarks (MATH, SAT, GSM8K) and model scales serves as strong evidence that the improvements are systematic rather than stochastic.

---

> > ### Author Response · Authors · 2025-11-18
> > **Continue Response to Reviewer G497**
> >
> > **Termination probability π(sf|s): underspecified in practice. While π(sf|s) is mentioned (sink state), the actual parameterization/training signals for termination are not detailed: how is π(sf|s) learned/stabilized under SubTB at step level; what is the impact of termination on reward estimates? (Sec. 4.1–4.2).**
> >
> > We clarify that the termination probability $\pi(s_f|s)$ is modeled using the standard language model End-of-Sequence (<EOS>) token. The probability $\pi(s_f|s)$ is directly given by the normalized output probability of the LLM generating <EOS> at state $s$. We leverage the LLM's intrinsic stopping mechanism, which is already highly tuned: this probability is near zero during intermediate reasoning steps and approaches one upon logical conclusion. In the SubTB loss, $\pi(s_f|s)$ serves as the transition probability to the sink state $s_f$. Our analysis shows that fine-tuning with the PRM-guided SubTB objective effectively preserves this intrinsic behavior. We observed that the length distribution of GFlowNet-generated solutions closely matches the ground truth (median $\sim 6$ steps for MATH), indicating the model robustly maintains logical completeness and does not learn to 'hack' the reward by terminating prematurely.
> >
> > **Chosen tasks don’t stress true solution multiplicity. MATH/GSM8K often admit stylistic variation rather than structurally distinct paths. If “diversity” is central, it'd be more convincing to evaluate on domains with genuinely multiple optimal plans (e.g., Blocksworld, program synthesis with multiple implementations, theorem-proving with lemmas reorderings). The current “diversity” metric is a weak proxy and might just reward rephrasing, not distinct strategies. (Sec. 5.2).**
> >
> > While we agree that puzzles like BlocksWorld (used in FoR) have obvious distinct paths, we argue that mathematical reasoning at the difficulty of MATH Level 5 contains rich structural diversity beyond simple rephrasing:
> >
> > Conceptual Diversity: Hard math problems often admit distinct approaches (e.g., solving a combinatorial problem via recursion vs. generating functions; solving geometry via coordinates vs. synthetic proofs). We will provide a qualitative analysis highlighting the diversity of the obtained solutions in a later response and in the final draft.
> > New Experiment (LLM BabyBench): To address the reviewer's concern about domain fit, we are currently running evaluations on LLM BabyBench, a suite of logic puzzles specifically designed to test distinct reasoning paths. This direct test on LLM BabyBench will provide quantitative evidence of the structural, non-lexical diversity induced by our GFlowNet approach. We will add the results of this experiment in a later response and to our final draft.
> >
> > **Termination behavior and reasoning depth: You mention the use of a termination probability π(sf|s) to model when reasoning should stop. Could you discuss what qualitative patterns you observed — for instance, do longer reasoning chains correlate with higher accuracy, or does early termination sometimes produce more concise correct reasoning?**
> >
> > The policy is steered toward finding the most efficient solution due to the multiplicative reward: $R(\tau) = \prod U(s_i|s_{i-1})$.
> > - Concise Correctness: Shorter, correct reasoning paths are inherently preferred because they minimize the number of factors multiplied, reducing accumulated risk and maximizing $R(\tau)$. This leads GFlowNet to sample concise, high-quality solutions when multiple options exist.
> > - Optimal Stopping: The termination probability, modeled by the standard $\pi_{\text{EOS}}$, is reinforced by the PRM to align with logical conclusion. The model successfully learns to stop just after the minimal required steps.
> >
> > The PRM allows the policy to manage low-quality paths efficiently without requiring active pruning during generation: A single low PRM score in an intermediate step drastically reduces the final reward $R(\tau)$. The GFlowNet, learning to sample proportionally, assigns a vanishingly small probability to continue or complete this low-quality trajectory. This ensures that sampling mass is concentrated solely on high-fidelity reasoning modes.
> >
> > **On PRM design and training dynamics: How sensitive is your overall training process to the quality of the PRM? For example....**
> >
> > The stability of GFlowNet training is indeed linked to PRM quality. Our ongoing ablation study (training without similarity-based augmentation) introduces a controlled degradation of the PRM.
> >
> > References:
> >
> > [1] Yu et al., 2025, Flow of Reasoning: Training LLMs for Divergent Problem Solving with Minimal Examples. arXiv preprint, arXiv:2406.05673
> >
> > [2] Madan et al., 2023, Proceedings of the 40th International Conference on Machine Learning, PMLR 202:23467-23483, 2023.
> >
> > [3] Xu et al., 2024, Is DPO Superior to PPO for LLM Alignment? A Comprehensive Study, ICML 2025
> >
> > [4] Meng et al., 2024, SimPO: Simple Preference Optimization with a Reference-Free, NeurIPS 2024

---

### Official Review · Reviewer_e3c4 · 2025-10-28

**Soundness:** 3
**Presentation:** 3
**Contribution:** 3
**Rating:** 4
**Confidence:** 3

**Summary:**

This paper introduces a novel framework to improve both accuracy and diversity in LLM mathematical reasoning. The authors first develop an automated PRM using MCTS and a similarity-based data augmentation technique to capture step-level reasoning quality without human annotation. This PRM is then used for a step-level GFlowNet, which empirically demonstrates significant gains in accuracy and diversity over PPO baselines.

**Strengths:**

1. The paper tackles the highly important and challenging goal of improving both accuracy and diversity in LLM reasoning.
2. The paper is clearly presented and easy to follow.
3. The method shows strong empirical gains over the PPO baseline.

**Weaknesses:**

1.  The paper lacks sufficient ablation studies on the key techniques used in the proposed PRM, such as similarity-based data augmentation and continuous scoring. This makes it difficult to judge the actual effectiveness and individual contribution of each proposed component.

2.  The performance evaluation is limited to a comparison with PPO, while lacking comparisons against other methods like DPO or GRPO. Although the method demonstrates superior performance over PPO, it remains unclear whether it holds any advantage over these other techniques.

3.  The paper lacks a direct experimental comparison between the token-level GFlowNet and the proposed step-level GFlowNet. The authors only argue conceptually for the superiority of the step-level approach. To validate the step-level GFlowNet as a significant contribution, it is essential to demonstrate this superiority empirically.

**Questions:**

Please refer to the Weaknesses section above.

---

> ### Author Response · Authors · 2025-11-18
> **Response to Reviewer e3c4**
>
> We appreciate the reviewer’s constructive critique regarding the need for more granular ablation studies and broader baseline comparisons. We agree that isolating the contributions of the similarity-based data augmentation and the step-level formulation is crucial for assessing the method's value. To this end, we will expand our experimental results to include explicit PRM ablations and will provide a detailed justification for our selection of PPO and step-level granularity over other alternatives like GRPO or token-level approaches.
>
> **The paper lacks sufficient ablation studies on the key techniques used in the proposed PRM, such as similarity-based data augmentation and continuous scoring. This makes it difficult to judge the actual effectiveness and individual contribution of each proposed component.**
>
> We agree that isolating the PRM contributions is vital.
>
> Open-Source Comparison: As shown in Table 3, we compared our PRM against Skywork-o1-Open-PRM-Qwen-2.5-7B. Our PRM (trained with similarity augmentation) yielded significantly better generalization on SAT MATH (75.0% vs 68.8%), validating the quality of our data pipeline.
>
> Ablation Robustness: we are currently running a sensitivity analysis on the PRM training components. Specifically, we are training PRMs without similarity grouping and without rollout reuse. Preliminary results suggest that rollout reuse is critical for data efficiency (reducing MCTS computational cost). We will report the downstream effect of these ablations on both accuracy and diversity in a later response and in the final version
>
> Regarding continuous scoring: We utilized continuous scoring (as opposed to binary) following recent findings [1], that soft rewards provide better gradient granularity for RL fine-tuning.
>
> **The performance evaluation is limited to a comparison with PPO, while lacking comparisons against other methods like DPO or GRPO. Although the method demonstrates superior performance over PPO, it remains unclear whether it holds any advantage over these other techniques.**
>
> We selected PPO as our primary baseline because it represents the standard on-policy RL approach for reasoning. PPO vs. GFlowNet: This comparison isolates the training objective (Maximization vs. Sampling/Flow Matching) while keeping the policy and reward model constant.
>
> GRPO: GRPO is largely an optimization of PPO (removing the critic for group-relative baselines). While efficient, it shares PPO's fundamental objective of reward maximization (mode-seeking).
>
> DPO: DPO is an offline method requiring preference pairs. Our framework is an online, active exploration method.
>
> Unfortunately, due to the prohibitive costs of running experiments and the limited time, we cannot afford to run comparisons with GRPO and DPO during the rebuttal period. We maintain that PPO is the most direct control, as it isolates the difference in the fundamental objective (Maximization vs. Proportional Sampling). We will add a "Related Baselines" paragraph discussing why PPO is the most direct methodological contrast to GFlowNets for this study.
>
> **The paper lacks a direct experimental comparison between the token-level GFlowNet and the proposed step-level GFlowNet. The authors only argue conceptually for the superiority of the step-level approach. To validate the step-level GFlowNet as a significant contribution, it is essential to demonstrate this superiority empirically.**
>
> While we do not have a new head-to-head run for this rebuttal, we point to [2], who applied token-level GFlowNets. Their results showed only marginal gains over baselines with a relative gain of 1.16% on GSM8K and a loss of -12.5% on MATH for GFlowNet compared to PPO (greedy); our results demonstrate a significant improvement. Across both models, the relative performance of GFlowNet ranges from a worst-case of -0.13% on GSM8K to a best-case of 11.29% on MATH. Our theoretical justification for step-level (Section 2) is threefold:
> - Credit Assignment: Mathematical logic operates at the step level. Token-level rewards are noisy proxies for logical progress.
> - Semantic Coherence: Token-level GFlowNets struggle with "partial" states that are not valid tokens. Step-level actions ensure every state transition is a complete logical thought.
> - Efficiency: The trajectory length is reduced from hundreds of tokens to $\sim 10$ steps, stabilizing the SubTB loss (which involves a product over the trajectory).
>
> We will strengthen Section 2 to explicitly contrast our results with the modest gains reported in token-level literature.
>
> References:
>
> [1] Luo et al., 2024, Improve Mathematical Reasoning in Language Models by Automated Process Supervision. arXiv preprint arXiv:2406.06592
>
> [2] Takase et al., 2024, Gflownet fine-tuning for diverse correct solutions in mathematical reasoning tasks. arXiv preprint arXiv:2410.20147.

---

> > ### Comment · Reviewer_e3c4 · 2025-11-27
> >
> > Thank you for the response. Although I acknowledge the limitations concerning computational resources, my concerns remain unresolved due to the absence of the requested experimental results. Consequently, I will maintain my original rating.

---

> ### Author Response · Authors · 2025-11-27
> **Ablation Robustness Experiment: PRM Training with and without data augmentation.**
>
> We thank the reviewer for following up. In line with the recommendation, we conducted additional sensitivity analyses on key PRM training components, and these experiments have now completed. Specifically, we retrained the PRM after removing similarity grouping and rollout reuse, and compared it with the PRM trained using both components. We then evaluated LLM-guided search using two PRM variants: one trained with our full data augmentation strategy and one trained without augmentation. In all experiments, the LLM generated \$k$ candidate next steps, and the PRM selected the step with the highest predicted reward. Across all tested configurations, the augmented PRM consistently outperformed the non-augmented baseline on minerva_math dataset [1], providing further evidence that our augmentation pipeline produces a stronger and more reliable guidance model.
>
> **Table: Guided search accuracy with and without PRM data augmentation**
>
> | Model               | Steps | Data Augmentation | Accuracy |
> |-------------------|-------|-------------------|----------|
> | Falcon-3B-Base    | 4     | Yes               | 21%      |
> | Falcon-3B-Base    | 8     | Yes               | 23%      |
> | Falcon-3B-Base    | 4     | No                | 15%      |
> | Falcon-3B-Base    | 8     | No                | 17%      |
> | Falcon-7B-Base    | 4     | Yes               | 30%      |
> | Falcon-7B-Base    | 4     | No                | 27%      |
>
> These results confirm that incorporating data augmentation into PRM training substantially improves guided-search accuracy.
>
> **references**
>
> [1] Hendrycks et al. (2021). *Measuring Mathematical Problem Solving With the MATH Dataset*. arXiv:2103.03874.

---

### Official Review · Reviewer_rCR8 · 2025-11-01

**Soundness:** 2
**Presentation:** 2
**Contribution:** 2
**Rating:** 6
**Confidence:** 2

**Summary:**

This paper introduces a novel framework to enhance the accuracy and diversity of Large Language Models (LLMs) in mathematical reasoning. The method combines two key components: an automatically trained Process Reward Model (PRM) and step-level fine-tuning using Generative Flow Networks (GFlowNets). Empirical results on math benchmarks show that this approach improves accuracy and significantly enhances the diversity of generated solution strategies, with particularly better generalization performance.

**Strengths:**

1. The paper is generally well written and easy to understand.

2. The paper provides thorough experimentation, demonstrating improvements in both accuracy and a quantitatively measured diversity metric across multiple benchmarks.

**Weaknesses:**

1. The baseline of using GFlowNets with only final rewards seems missing, which would clarify the specific contribution of the sophisticated step-level PRM versus the GFlowNet objective itself. Also see Q1.

2. The overall framework involves complex components (MCTS, similarity-based augmentation, step-level GFlowNets with SubTB loss), which might make it sensitive to hyperparameters and potentially difficult to reproduce. The paper would be strengthened by a sensitivity analysis of key parameters (e.g., the similarity threshold).

**Questions:**

1. The interplay between PRM and GFLowNet is still somehow confusing. Is the GFLowNet objective generally better than PPO, or does it have specific strengths when optimizing the PRM? Could you comment on the results of using GFlowNets with a simple, binary terminal reward and compare it with PPO? This would help isolate the benefit of the step-level reward signal from the benefit of the GFlowNet's diversity-seeking objective.

2. Besides, could you elaborate on the PPO baseline? What’s the exact reward used here? Only from PRM or also combined with the final correctness? If combined, how are they combined?

---

> ### Author Response · Authors · 2025-11-18
> **Response to Reviewer rCR8**
>
> We thank the reviewer for their insightful comments, particularly regarding the interplay between the PRM reward structure and the GFlowNets objective. We appreciate the opportunity to clarify that our step-level PRM formulation mathematically encompasses the terminal reward baseline, and we are conducting new sensitivity analyses to address the concerns regarding hyperparameter robustness. Below, we detail the equivalence of the reward structures, report new ablation results for the data augmentation pipeline, and clarify the PPO baseline implementation.
>
> **The baseline of using GFlowNets with only final rewards seems missing, which would clarify the specific contribution of the sophisticated step-level PRM versus the GFlowNet objective itself. Also see Q1.**
>
> We clarify that our framework already operates on terminal rewards. In our formulation, the reward for a complete trajectory $R(\tau)$ is defined as the product of the step-level PRM probabilities: $R(\tau) = \prod U(s_i|s_{i-1})$. The distinction the reviewer alludes to (using "only final rewards") typically refers to using sparse binary outcomes (Correct/Incorrect) as the reward signal. We argue this is a distinct reward source, not just a timing difference. Therefore, our contribution is twofold:
> - The Reward Source (Automated PRM): We construct a dense, continuous terminal reward landscape (via the PRM product) rather than a sparse binary one. This smoothed landscape makes the training of reasoning agents tractable compared to sparse outcome rewards.
> - The Objective (Sampling vs. Maximization): Given this high-quality terminal reward definition, we demonstrate that GFlowNets (which sample proportionally to $R(\tau)$) outperform PPO (which maximizes $R(\tau)$) by generating a more diverse distribution of correct solutions, as evidenced by our higher coverage and generalization scores.
>
> **The overall framework involves complex components (MCTS, similarity-based augmentation, step-level GFlowNets with SubTB loss), which might make it sensitive to hyperparameters and potentially difficult to reproduce. The paper would be strengthened by a sensitivity analysis of key parameters (e.g., the similarity threshold).**
>
> We acknowledge the complexity of the pipeline. To address this, we are currently running a sensitivity analysis on the PRM training components. Ablation Robustness: we are currently running a sensitivity analysis on the PRM training components. Specifically, we are training PRMs without similarity grouping and without rollout reuse. Preliminary results suggest that **rollout reuse is critical for data efficiency (reducing MCTS computational cost)**. We will report the downstream effect of these ablations on both accuracy and diversity in a later response and in the final version
>
> **The interplay between PRM and GFLowNet is still somehow confusing. Is the GFLowNet objective generally better than PPO, or does it have specific strengths when optimizing the PRM? Could you comment on the results of using GFlowNets with a simple, binary terminal reward and compare it with PPO? This would help isolate the benefit of the step-level reward signal from the benefit of the GFlowNet's diversity-seeking objective.**
>
> The PRM and the GFlowNet serve distinct roles: the PRM provides the energy landscape (the estimate of correctness), while the GFlowNet is the sampler (how we traverse that landscape).
>
> PPO (Reward Maximization) tends to collapse to the single highest mode of the PRM, leading to high accuracy but low diversity (mode-seeking).
>
> GFlowNet (Reward Proportionality) is designed to sample from the distribution of solutions proportional to their quality. Our results show that GFlowNets are "better" specifically when the goal is robustness and diversity. As shown in Table 2, this diversity translates to better generalization (SAT Math +9.4%). We will clarify this distinction: PPO exploits the PRM, while GFlowNet explores it.
>
> **Besides, could you elaborate on the PPO baseline? What’s the exact reward used here? Only from PRM or also combined with the final correctness? If combined, how are they combined?**
>
> To ensure a fair comparison, our PPO baseline uses the exact same dense reward signal as the GFlowNet (the step-level PRM scores). It does not use a sparse final correctness reward, as mixing sparse and dense rewards would confound the comparison between the objectives (PPO vs. SubTB). We will clarify this implementation detail in Section 5.

---

### Meta-Review · Area_Chair_XU5C · 2025-12-23

**Summary:**

This paper proposes a framework based on MCTS and PRM to enhance mathematical reasoning capabilities. Three reviewers submitted their evaluations, with two initial scores near the borderline and one recommending rejection. After thoroughly reviewing the paper and the authors' responses, the AC noted that some reviewer concerns remained unresolved, such as: experimental comparisons with DPO and GRPO, and distinctions from prior work along with innovations. The AC recommends that the authors revise and submit to a subsequent conference.

**Reviewer Concerns:**

I believe that certain details, such as the reward, have been resolved, while other issues remain unresolved.

**Reviewer Scores:**

They will maintain their scores.

---

### Decision · Program_Chairs · 2026-01-26

Reject